# Steroids and Fatty Acid Esters from *Cyperus sexangularis* Leaf and Their Antioxidant, Anti-Inflammatory and Anti-Elastase Properties

**DOI:** 10.3390/molecules28083434

**Published:** 2023-04-13

**Authors:** Gugulethu Mathews Miya, Ayodeji Oluwabunmi Oriola, Bianca Payne, Marizé Cuyler, Namrita Lall, Adebola Omowunmi Oyedeji

**Affiliations:** 1Department of Chemical and Physical Sciences, Faculty of Natural Sciences, Walter Sisulu University, Mthatha 5117, South Africa; aoyedeji@wsu.ac.za; 2Department of Plant and Soil Sciences, University of Pretoria, Pretoria 0002, South Africa; u12179711@up.ac.za (B.P.); marizenel7@gmail.com (M.C.); namrita.lall@up.ac.za (N.L.)

**Keywords:** *Cyperus sexangularis*, antioxidant, anti-inflammatory, anti-elastase, steroids, fatty acid esters

## Abstract

*Cyperus sexangularis* (CS) is a plant in the sedges family (Cyperaceae) that grows abundantly in swampy areas. The leaf sheath of plants in the Cyperus genus are mostly used domestically for mat making, while they are implicated for skin treatment in traditional medicine. The plant was investigated for its phytochemical contents as well as its antioxidant, anti-inflammatory and anti-elastase properties. The n-hexane and dichloromethane leaf extracts were chromatographed on a silica gel column to afford compounds **1**–**6**. The compounds were characterized by nuclear magnetic resonance spectroscopy and mass spectrometry. The inhibitory effect of each compound against 2,2-diphenyl-1-picrylhydrazyl (DPPH), nitric oxide (NO) and ferric ion radicals were determined by standard in vitro antioxidant methods. The in vitro anti-inflammatory response was measured using egg albumin denaturation (EAD) assay, while the anti-elastase activity of each compound in human keratinocyte (HaCaT) cells was also monitored. The compounds were characterized as three steroidal derivatives, stigmasterol (**1**), 17-(1-methyl-allyl)-hexadecahydro-cyclopenta[a]phenanthrene (**2**) and β-sitosterol (**3**), dodecanoic acid (**4**) and two fatty acid esters, ethyl nonadecanoate (**5**) and ethyl stearate (**6**). Stigmasterol (**1**) exhibited the best biological properties, with IC_50_ of 38.18 ± 2.30 µg/mL against DPPH, 68.56 ± 4.03 µg/mL against NO and 303.58 ± 10.33 µAAE/mg against Fe^3+^. At 6.25 µg/mL, stigmasterol inhibited EAD by 50%. This activity was lower when compared to diclofenac (standard), which demonstrated 75% inhibition of the protein at the same concentration. Compounds **1**, **3**, **4** and **5** showed comparable anti-elastase activity with an IC_50_ ≥ 50 µg/mL, whereas the activity of ursolic acid (standard) was double fold with an IC_50_ of 24.80 ± 2.60 µg/mL when compared to each of the compounds. In conclusion, this study has identified three steroids (**1**–**3**), one fatty acid (**4**), and two fatty acid esters (**5** and **6**) in *C. sexangularis* leaf for the first time. The compounds showed considerable antioxidant, anti-inflammatory and anti-elastase properties. Thus, the findings may serve as a justification for the folkloric use of the plant as a local skin ingredient. It may also serve to validate the biological role of steroids and fatty acid compounds in cosmeceutical formulations.

## 1. Introduction

Skin illnesses are prevalent in all human age groups and may be due to exposure towards radiation, biological toxins, bacteria, natural chemical agents found in the environment and, to some extent, due to starvation [1]. Skin also gets dehydrated when it is exposed to extreme heat during summer, and this causes sunburns, marks, pigment, wrinkles, and spots [2]. Excessive coldness which usually occurs in winter can also cause severe harm to the skin in the form of cracks, cuts, laceration, and infections [1].

Oxidative stress is one of the main mechanisms that leads to skin aging and other skin problems, while sunlight ultraviolet (UV) radiation is one of the most known harmful factors that causes skin problems [3]. Alteration in skin tissues due to formation of reactive oxygen species and lipid peroxides also leads to skin aging and a whole lot of skin problems [4]. The skin-generated reactive oxygen species react with several biomolecules to cause a noticeable alteration in the normal skin conditions and a continued inflammatory process [5].

The dermal layer is structurally a flexible and reliable component of the skin [6]. Elastin within the dermal layer is one of the structural components that makes a massive contribution in tissues and tendons connection in humans because of the insolubility of elastic fibrous protein [7]. Elastin is an important part of numerous human body tissues, such as lung, skin, and arteries [8]. Due to their dependence in elasticity, elastin’s presence in these tissues gives them the ability to stretch and recoil; thus, it is the major component in skin elasticity and firmness [8]. The main problem is, with ageing, elastin decreases drastically, which results in loss of skin flexibility and strength; thus, visible wrinkles materialize due to the low elastin production rate as one is ageing [9,10]. The decreased quality of elastin due to elastase–elastin interaction has resulted in greater interest in degradative enzymes such as elastase because of its role in skin ageing [11].

Skin normally requires elastase to remove non-functional proteins that remain in the extracellular matrix; for example, skin wound portion reconstruction needs dysfunctional proteins to be removed before the repair can occur [12,13]. An increase in elastase activity has been reported to act as a trigger to various inflammatory responses involving proteinases, which can eventually lead to the destruction of elastin and collagen in the skin [14]. The required cells for skin reconstruction also decrease due to ageing [11]. Based on the above reasons, sustainable inhibitors that can inhibit elastase enzymes can be employed in the treatment of wrinkles to fight skin ageing appearance [15]. Therefore, the application of antioxidants, anti-inflammatory and elastase inhibitors can provide a resourceful strategy to decrease oxidative damage and increase elastin production, thus delaying the skin ageing condition [16].

In recent times, more attention has been paid to the application of herbal cosmetics when compared to drug/synthetic cosmetics, because the herbal cosmetic ingredients have been found to be more potent and biocompatible than the synthetic ones, with little or no side effects [17]. As a matter of fact, some synthetic skin care ingredients, such as fragrances, dyes, propylene glycol, parabens, synthetic colorants, and sodium lauryl sulfate, have been reported to show some level of adverse reactions [18,19].

*Cyperus sexangularis* Nees (Family Cyperaceae) is a tufted, long-lasting (perennial) sedge that thrives in the wetland habitat [20]. It has a clump-forming shape, grows fast, is always moist, is very hardy and evergreen, flowers in summer with red flowers and can grow up to 1.5 m high [20]. The plant has been implicated in traditional medicines for the management of inflammatory diseases, including skin-related problems, while other medicinal uses include management of menstrual irregularities, blood disorders, diarrhea, and dysentery [21]. Compounds such as stigmasterol, β-amyrin acetate, 4-hydroxy butyl cinnamate, β-sitosterol, lupeol, gallic acid, quercetin, β-amyrin, oleanolic acid, 4-hydroxy cinnamic acid, kaempferol and caffeic acid have been reported in other *Cyperus* species in addition to their essential oil rich contents such as β-himachalene, α-humulene, γ-himachalene, cyperene, p-vinylguaiacol, copaene, limonene, cymene, pinene, 4-terpineol and (−)-cyperene [22,23,24].

Currently, there is a dearth of information on the chemical constituents and biological potentials of *C. sexangularis*. Therefore, the study was carried out to isolate and identify some of the chemical constituents of *C. sexangularis* leaf, as well as evaluate the antioxidant, anti-inflammatory and anti-elastase properties of the compounds.

## 2. Results and Discussion

### 2.1. Spectral Data

The structure elucidation of compounds **1**–**6** was based on their acquired (observed) spectra, which are presented in Appendix A.

Compound **1** (27 mg): stigmasta-5,22-dien-3-ol; C_29_H_48_O (exact mass = 412.69 g/mol); **TOF ES+/MS** (% rel. abundance): *m*/*z* 435.3662 (38.5%) [C_29_H_48_O+Na]^+^, *m*/*z* 355.2563 (40.0%) [M-57]^+^, *m*/*z* 204.3010 (88.0%) [M-108]^+^, *m*/*z* 157.0350 (M+, 100%) [M-255]^+^; **^1^H NMR** (600 MHz in CDCl_3_), δ ppm: 0.68 (3H, s, H-19), 0.80 (3H, d, *J* = 6.9 Hz, H-29), 0.82 (3H, d, *J* = 6.9 Hz, H-28), 0.84 (3H, t, *J* = 8.1 Hz, H-26), 0.92 (3H, d, *J* = 6.7 Hz, H-21), 1.00 (3H, s, H-18), 3.53 (1H, dt, *J* = 6.0 Hz, H-3), 5.35 (1H, bd, *J* = 6.0 Hz, H-6); **COSY**: 2.25 (H-4) and 3.53 (H-3), 5.01 (1H, dd, H-23) and 5.15 (1H, dd, H-22); **^13^C NMR** (150 MHz in CDCl_3_), δ ppm: 37.40 (C-1), 34.08 (C-2), 71.95 (C-3), 42.42 (C-4), 140.89 (C-5), 121.87 (C-6), 31.78 (C-7), 32.04 (C-8), 50.27 (C-9), 36.29 (C-10), 21.23 (C-11), 39.92 (C-12), 42.45 (C-13), 56.91 (C-14), 24.45 (C-15), 29.29 (C-16), 56.20 (C-17), 12.12 (C-18), 18.93 (C-19), 40.63 (C-20), 19.97 (C-21), 138.46 (C-22), 129.41 (C-23), 45.97 (C-24), 26.21 (C-25), 12.00 (C-26), 29.85 (C-27), 19.54 (C-28), 19.18 (C-29). The full NMR assignments (Table 1) were compared to literature reports on stigmasterol [24,25,26].

Compound **2** (21 mg): 17-(1-methyl-allyl)-hexadecahydro-cyclopenta[a] phenanthrene; C_21_H_34_ (exact mass = 286.15 g/mol); ^1^**H NMR** (400 MHz in CDCl_3_), δ ppm: 0.87 (3H, d, *J* = 4.0 Hz, H-19), 4.96 (2H, dd, *J* = 16.0, 8.0 Hz, H-21), 5.82 (1H, m, H-20); **^13^C NMR** (100 MHz in CDCl3), δ ppm: 28.74 (C-1), 26.45 (C-2), 24.18 (C-3), 29.40 (C-4), 31.70 (C-5), 29.07 (C-6), 25.83 (C-7), 31.39 (C-8), 29.74 (C-9), 32.33 (C-10), 28.42 (C-11), 26.79 (C-12), 32.96 (C-13), 33.36 (C-14), 22.24 (C-15), 19.26 (C-16), 36.88 (C-17), 32.02 (C-18), 13.81 (C-19), 138.72 (C-20), 113.65 (C-21). The full NMR assignments (Table 2) were compared to 5β-pregnane reported by Cyril-Olutayo et al. [27]. 

Compound **3** (10 mg): β-sitosterol; C_29_H_50_O (exact mass = 414.71 g/mol); **^1^H NMR** (400 MHz in C_6_D_6_), δ ppm: 0.66 (3H, s, H-19), 0.78 (3H, d, *J* = 4.0 Hz, H-29), 0.80 (3H, d, *J* = 4.0 Hz, H-28), 0.82 (3H, t, *J* = 4.5, H-26), 0.91 (3H, d, *J* = 6.4 Hz, H-21), 0.99 (3H, s, H-18), 2.25 (2H, d, *J* = 6.8 Hz, H-4), 3.52 (1H, tdd, J = 9.6, 6.4, 6.0 Hz, H-3), 5.32 (1H, bd, J = 5.2 Hz, H-6); **^13^C NMR** (100 MHz in CDCl_3_), δ ppm: 37.48 (C-1), 31.86 (C-2), 71.78 (C-3), 42.28 (C-4), 140.66 (C-5), 121.39 (C-6), 30.36 (C-7), 29.73 (C-8), 50.02 (C-9), 36.86 (C-10), 22.93 (C-11), 39.52 (C-12), 42.28 (C-13), 56.60 (C-14), 28.09 (C-15), 29.73 (C-16), 55.82 (C-17), 11.84 (C-18), 17.98 (C-19), 36.24 (C-20), 19.23 (C-21), 35.61 (C-22), 25.74 (C-23), 45.71 (C-24), 24.33 (C-25), 11.96 (C-26), 29.69 (C-27), 20.88 (C-28), 19.28 (C-29). The full NMR assignments (Table 1) were compared to literature reports [24,25,26].

Compound **4** (22 mg): dodecanoic acid; C_12_H_24_O_2_ (exact mass = 200.32 g/mol); ^1^**H NMR** (400 MHz in C_6_D_6_), δ ppm: 0.86 (3H, t, *J* = 8.0, 8.0 Hz, H-12), 1.23–1.28 (2H, m, H-3 to H-10), 1.58 (2H, t, *J* = 9.6, 7.6 Hz, H-2); **^13^C NMR** (150 MHz in C_6_D_6_/TFT), δ ppm: 179.84 (C-1), 31.92 (C-2), 29.36 (C-3), 29.85 (C-4/C-5/C-6/C-7/C-8, C-9), 29.89 (C-10), 22.89 (C-11), 14.12 (C-12). The full NMR assignments (Table 3) were compared to lauric acid and undecanoic acid [28,29].

Compound **5** (19 mg): ethyl nonadecanoate; C_21_H_42_O_2_ (exact mass = 326.56 g/mol); **TOF ES+/MS** (% rel. abundance): *m*/*z* 327.2211 (19.0%) [M+H]^+^, *m*/*z* 290.1702 (38.5%), *m*/*z* 157.0352 (100%, M^+^) [M-169]^+^; **^1^H NMR** (600 MHz in C_6_D_6_), δ ppm: 0.91 (3H, t, *J* = 6.9 Hz, H-19), 1.26 (3H, t, *J* = 16.4 Hz, H-2′), 2.21 (2H, t, *J* = 7.4 Hz, H-2), 4.07 (2H, t, *J* = 6.7 Hz, H-1′); **COSY**: 1.61 (H-3) and 2.21 (H-2), 1.26 (H-18) and 0.91 (H-19); **^13^C NMR** (150 MHz in C_6_D_6_/TFT), δ ppm: 173.01 (C-1), 34.59 (C-2), 25.49 (C-3), 26.41 (C-4), 29.27 (C-5), 29.60 (C-6), 29.69 (C-7), 29.74 (C-8), 29.95 (C-9), 29.81 (C-10), 30.05 (C-11), 30.00 (C-12), 30.13 (C-13), 30.09 (C-14), 30.15 (C-15), 30.21 (C-16), 32.36 (C-17), 23.10 (C-18), 14.00 (C-19), 64.31 (C-1′), 14.30 (C-2′).

Compound **6** (21 mg)*:* ethyl stearate; C_20_H_40_O_2_ (exact mass = 312.51 g/mol); **TOF ES+/MS** (% rel. abundance): *m*/*z* 313.4124 (16%) [M+H]^+^, *m*/*z* 273.1667 (70.0%) [M-39]^+^, *m*/*z* 217.0994 (34.5%) [M-95]^+^, 157.0376 (M^+^, 100%) [M-155]^+^. **^1^H NMR** (600 MHz in C_6_D_6_), δ ppm: 0.91 (3H, t, *J* = 6.8 Hz, H-18), 1.26 (3H, t, *J* = 21.6 Hz, H-2′), 2.21 (2H, t, *J* = 7.4 Hz, H-2), 4.08 (2H, dd, *J* = 6.7, 6.7 Hz, H-1′); COSY: 0.91 (H-18) and 1.24 (H-17), 1.26 (H-2′) and 4.08 (H-1′); **^13^C NMR** (150 MHz in C_6_D_6_), δ ppm: 173.01 (C-1), 34.59 (C-2), 25.49 (C-3), 26.41 (C-4), 29.27 (C-5), 29.59 (C-6), 29.69 (C-7), 29.73 (C-8), 29.95 (C-9), 29.82 (C-10), 30.05 (C-11), 29.99 (C-12), 30.14 (C-13), 30.08 (C-14), 30.20 (C-15), 32.35 (C-16), 23.10 (C-17), 14.29 (C-18), 64.31 (C-1′), 14.29 (C-2′). The full NMR assignments of compounds 5 and 6 (Table 4) were compared to literature reports on ethyl stearate [30].

The structures of the six compounds isolated from the leaf of *C. sexangularis* are presented in Figure 1. Stigmasterol (**1**), 17-(1-methyl-allyl)-hexadecahydro-cyclopenta[a]phenanthrene (**2**), β-sitosterol (**3**), dodecanoic acid (**4**), ethyl nonadecanoate (**5**) and ethyl stearate (**6**) are reported in this plant for the first time.

The observed NMR spectral data of compound **4** and, in comparison, with the spectra data of undecanoic acid (Lauric acid) reported by Abe et al. [31] as presented in Table 3, as well as that of decanoic acid reported by Sanguanphun et al. [30]. Therefore, the compound has been characterized as dodecanoic acid, a medium-chain fatty acid from the DCM extract of *C. sexangularis.*

### 2.2. Evaluation of the Biological Activities of Compounds Isolated from Cyperus sexangularis Leaf

The biological activities of plant extracts were based on their ability to inhibit the radical actions of DPPH and NO, reduce the ferric ion to ferrous ion, exhibit anti-inflammatory response by inhibiting protein denaturation and inhibit the activity of elastase enzyme in the human keratinocyte (HaCaT) cells, all in an in vitro milieu. Based on the antioxidant study, stigmasterol (1) was the most antioxidant (AOX) active compound (Table 5). It significantly reduced Fe^3+^ radicals at 303.58 ± 10.33 µgAAE/mg. Stigmasterol also inhibited DPPH and NO radicals at IC_50_ of 38.18 ± 2.30 µg/mL and 68.56 ± 4.03, respectively, and these activities were comparable (*p* > 0.05) to those observed for dodecanoic acid (**4**). However, L-ascorbic acid, the standard antioxidant compound, demonstrated about three- to five-fold more AOX strength when compared to the isolated compounds. A report has shown the ability of β-sitosterol to modulate antioxidant enzyme response in RAW 264.7 macrophages [32]. Phytosterols such as stigmasterol and β-sitosterol, which were isolated and potentiated in this study, have also been shown to chemically act as radical scavengers, and physically as a stabilizer in the membranes [33]. Some fatty acid methyl esters have also been reported to demonstrate optimal (>50%) DPPH radical scavenging activities at 100–10 µg/mL [34].

The inhibitory effects of the isolated compounds against protein (egg albumin) denaturation were presented in Figure 2. At the lowest concentration (6.25 µg/mL), stigmasterol (**1**) exhibited the highest (51.03 ± 10.36%) inhibition among the compounds. Moreover, it demonstrated up to 70% inhibition at a median (25 µg/mL) concentration. However, the anti-inflammatory response of 1 was significantly lower than that of diclofenac (standard), which demonstrated 75.63 ± 3.35% and 89.30 ± 0.12% at the lowest and median concentrations, respectively. The inhibitory effects of compounds 3–5 against egg albumin denaturation (EAD) were within the 80% and 70% marks when tested at 100 and 50 µg/mL concentrations, respectively, while ethyl stearate (6) showed the least activity among the fatty acid compounds at concentrations above 6.25 µg/mL. Phenanthrene derivative (**2**) was the least active among all the compounds tested, with significantly (*p* < 0.05) reduced inhibitory effects against EAD across all concentrations. Studies have shown the anti-inflammatory effects of phytosterols such as stigmasterol and β-sitosterol [35,36].

The anti-skin ageing property of the isolated compounds was monitored by their inhibition of the activity of elastase enzyme in human keratinocyte (HaCaT) cells, and the result is as presented in Table 6. The result showed that purification of the n-hexane and DCM extracts caused significant improvement in the anti-elastase activity of the plant. It was observed that the two sterols stigmasterol (**1**) and β-sitosterol (**3**), dodecanoic acid (**4**), ethyl nonadecanoate and ethyl stearate all demonstrated > 50 µg/mL anti-elastase activity, while phenanthrene derivative (**2**) exhibited > 100 µg/mL. It is noteworthy, that ursolic acid, a standard anti-elastase steroidal compound, exhibited a double-fold (24.80 ± 2.60 µg/mL) anti-elastase activity. Evidence has shown the marked anti-elastase activities of some steroidal compounds, including β-sitosterol, and fatty acid compounds with IC_50_ values ranging from 1.9–4.4 μM [37]. Stigamsterol has also been reported to aid a repair of degenerated dermal and epidermal layers of the skin in an in vivo study [38]. Thus, the potentiated compounds have been shown through this study to have considerable bioactivity, with potential to enhance skin elasticity.

It is worth mentioning that some plant sterols such as β-sitosterol and stigmasterol, medium-to-long chain fatty acids and their esters which were isolated and identified for the first time in *C. sexangularis* have been shown by many studies to portend considerable biofunctionalities, such as antioxidant, anti-inflammatory, anti-ageing and cytotoxic activities; thus, this may in part be the rationale for their cosmeceutical applications [39,40,41,42,43]. For instance, stearic acid ester and its derivatives are used nowadays as part of the ingredients in skincare products [44,45]. Therefore, the compounds isolated from this study may provide new insights for the discovery and/or development of more bioactive agents for cosmeceutical applications. 

## 3. Materials and Methods

### 3.1. General Experimental Procedure

All reagents, solvents, silica gel for TLC and silica gel for column chromatography were purchased from Sigma-Aldrich (Pty) Ltd. (Johannesburg, South Africa) and Merck (Pty) Ltd. (Johannesburg, South Africa) through a licenced local supplier, Shalom Laboratories and Supplies, South Africa. Antioxidant absorbance were measured on a 680-Bio-Rad Microplate Reader (Serial Number 14966, Irvine, CA, USA), while the anti-elastase activity was monitored on a Perkin Elmer VICTOR Nivo microplate reader (excitation: 560 nm/emission: 590 nm wavelengths). Isolated compounds were analyzed with a 600 MHz Bruker Biospin GmbH spectrometer (Bruker, Rheinstetten, Germany) and a 400 MHz VARIAN NMR spectrometer (Agilent Technologies Inc., Santa Clara, CA, USA). Chemical shift (δ) was recorded in part per million (ppm). The spectrum range was set at 0–14 ppm for 1H NMR, while that of ^13^C NMR was set at 0–220 ppm. Tetramethyl silane was used as the internal standard and compounds were read in deuterated chloroform and benzene, as well as benzene-d6 in trifluorotoluene (TFT). High-resolution mass spectrometry (HRMS) was performed with a Waters API Q-TOF Ultima spectrometer (Waters Corporation, Manchester, UK) situated in Stellenbosch University, Stellenbosch, South Africa.

### 3.2. Plant Material

Fresh leaves of *C. sexangularis* (CS) were collected from a Botanical Garden in the Limpopo Province. An herbarium specimen was deposited in the herbarium of the Department of Botany, University of Pretoria with voucher number MGM 008 for future reference. The leaves were then collected in large quantity, air-dried, and milled into powder for further work.

### 3.3. Gradient Extraction

The leaf powder of CS (723.90 g) was exhaustively extracted at room with 7.5 L each of n-hexane, dichloromethane (DCM), ethyl acetate (EtOAc) and ethanol (EtOH) to afford 11.62, 2.68, 1.07 and 9.48% yield, respectively.

### 3.4. Isolation of Compounds

#### 3.4.1. Column Chromatography of n-Hexane Extract of *C. sexangularis* (CS) Leaf

The n-hexane extract (80.20 g) was absorbed onto 50 g of silica gel (70-230 ASTM mesh, Merck KGaA, Darmstadt, Germany) and dry-packed with a 200 g silica gel as the separating layer. The column was eluted, using 700 mL each of gradient solvent systems of 10% increase in polarity from hexane (100%) to Hex-DCM (9:1) and Hex-DCM (8:2) up to DCM (100%), through DCM-EtOAc (9:1) up to EtOAc (100%), and finally with the introduction of MeOH in the following manner: EtOAc-MeOH (9:1, 1:1, 0:100). A total of 171 eluates (column fractions) were collected, and based on TLC analysis, fractions 1–2 gave a whitish crystalline deposit after 48 h of chromatographic fractionation. It was washed with hexane (100%) to afford compound 1, (R_f_ 0.48 in DCM 100%) fluoresced at 254 and 366 nm wavelengths under the UV light, appeared as a single pinkish spot when sprayed with 10% H_2_SO_4_, and gave a sharp melting point range of 164–165 °C, thus indicating compound 1 to be a pure steroidal or terpenoidal compound [46].

#### 3.4.2. Column Chromatography of DCM Extract of *C. sexangularis* (CS) Leaf

The dichloromethane extract (14.60 g) was adsorbed onto 60 g silica gel and dry-packed on a 200 g silica gel column. The adsorbed extract was gradiently eluted with 500 mL each of solvent system of increasing level of polarity, Hex-DCM (100:0, 9:1, 8:2, 7:3, 6:4, 1:1, 4:6, 3:7, 2:8, 1:9 and 0:100), DCM-EtOAc (100:0, 9:1, 8:2, 7:3, 6:4, 1:1, 4:6, 3:7, 2:8, 1:9 and 0:100) and EtOAc-MeOH (100:0, 9:1, 7:3, 0:100). The column afforded 149 eluates, which resulted in the four groups of solid deposits from 1–55. Based on TLC analysis, each group was washed with hexane (100%) to afford five compounds (**2**–**6**). Compound **2** was isolated as a whitish semisolid substance, with no UV fluorescence. It turned purplish on TLC analysis (Hex-DCM 9:1, R_f_ 0.42), to suggest it as a saturated terpene. Compound **3** was isolated as a whitish crystalline, UV active molecule. It gave a similar pinkish spot compared to **1**, but at an R_f_ value of 0.46 when it was developed in DCM 100% solvent system and a melting point range of 136–138 °C. Compounds **4**–**6** were isolated as a whitish semisolid, with purplish spot each at R_f_ 0.68, 0.75 and 0.74, respectively, in a Hex-EtOAc (7:3) solvent system.

### 3.5. Antioxidant Tests

#### 3.5.1. DPPH Spectrophotometric Assay

This was performed by following a previously reported conventional method [47], with some modifications. Here, 0.5 mL of 0.1Mm DPPH radical in methanol was added to 0.5 mL of serially diluted CS-isolated compounds and *L*-ascorbic acid at 100.00, 50.00, 25.00, 12.50 and 6.25 µg/mL concentration range in triplicate. The reaction mixture was incubated in the dark at 37 °C for 30 min. The absorbance was measured at 515 nm on a microplate reader. The percentage inhibition of the radical was calculated thus:$$\%\;DPPH\;inhibition=\left(\frac{ABScontrol-ABSsample}{ABScontrol}\right)\times 100$$
where *ABSsample* = Absorbance of test sample, while *ABScontrol* = Absorbance of negative control (2% DMSO).

#### 3.5.2. Nitric Oxide (NO) Inhibition Assay

The inhibitory effect of the isolated compounds against the NO radical was investigated using the method described by Jimoh et al. [48]. The compounds were prepared in varying concentrations from 100.00–6.25 µg/mL. They were added to 0.2 Mm (2 mL) of sodium nitroprusside in triplicates. The reaction mixture was incubated at 25 °C for 3 h, and thereafter, 0.5 mL of the mixture was mixed with Griess reagent (0.33% sulphanilamide dissolved in 20 % glacial acetic acid and mixed with 1 mL of naphthylethylenediamine chloride (0.1% *w*/*v*). The mixture of the complex and Griess reagent were incubated at room temperature for 30 min. This was immediately followed by absorbance measurement at 540 nm on a microplate reader. The IC_50_ of each test sample was also determined after the determination of their percentage inhibition, as thus:$$\%\;NO\;inhibition=\left(\frac{ABScontrol-ABSsample}{ABScontrol}\right)\times 100$$

#### 3.5.3. Ferric Reducing Antioxidant Power (FRAP) Assay

The reducing antioxidant capacity of each isolated compound was evaluated according to the method of Kumar and Pandey [49], with some modifications. The compounds were dissolved in 2% DMSO and added to the mixture containing 0.25 mL of phosphate buffer (0.2 M; pH 6.6) and 0.25 mL of potassium ferricyanide [K_3_Fe(CN)_6_] (1% *w*/*v*). Standard compounds used were *L* (+)-Ascorbic acid. The resulting mixture was incubated at 50 °C for 20 min, followed by the addition of 0.25 mL of CCl_3_COOH (10% *w*/*v*) and then centrifuged at 3000 rpm for 10 min. The upper layer of the solution was mixed with 1 mL of 2% DMSO and the same amount and 0.5 mL of FeCl_3_ (0.1%, *w*/*v*). The experiment was conducted in triplicate, and absorbance was measured at 700 nm on a microplate reader against a blank sample of only phosphate buffer. Thus, the ferric reducing power of the isolated compounds at 100.00, 50.00, 25.00, 12.50 and 6.25 µg/mL concentration in methanol were determined as ascorbic acid equivalent (AAE) from the calibration curve of the positive control (L-ascorbic acid).

### 3.6. In Vitro Anti-Inflammatory Test

The test was carried out using the egg albumin denaturation assay method reported by Chatterjee et al. [50]. Here, a reaction mixture comprising 0.2 mL of albumin content of fresh chicken egg, 2.8 mL of phosphate buffer saline (pH 6.4) and 2 mL each of the isolated compounds at varying concentrations (6.25–100 µM) was prepared in triplicate. The mixture was incubated at 37 °C for 15 min away from direct light and thereafter boiled at 70 °C for 5 min in a thermostatic water bath. The resulting mixture was cooled, and the absorbance was measured at 655 nm on a microplate reader. Diclofenac was similarly tested as the standard drug, and the percentage inhibition of the samples was calculated thus:$$\%\;inhibition\;of\;EAD=\left(\frac{ABScontrol-ABSsample}{ABScontrol}\right)\times 100$$
where *EAD* = egg albumin denaturation, *ABSsample* = absorbance of compounds/Diclofenac, *ABScontrol* = absorbance of negative control (vehicle), i.e., 2% DMSO).

### 3.7. Anti-Elastase Assay

#### 3.7.1. Cell Culture

The method as described by Lall et al. [51] was followed for the in vitro cytotoxicity assay. Human keratinocyte (HaCaT) cells were grown in Dulbecco’s Modified Eagle Medium (DMEDM) supplemented with 1% antibiotics containing penicillin (100 U/mL)-streptomycin (100 μg/mL), 1% fungizone (250 μg/L) and 10% fetal bovine serum. The cells were grown in T75 cell culture flasks humidified incubator and set at 5% CO_2_ and 37 °C until a confluent monolayer formed. Trypsin was used to detach and sub-culture cells once a confluent monolayer had formed.

#### 3.7.2. Test Procedure

The ability of each isolated compound to inhibit porcine pancreatic elastase (PPE) (Sigma) was determined by measuring the release of p-nitroaniline from N-succinyl-ala-ala-ala-p-nitroanilide spectrophotometrically at 405nm according to the method of Bieth et al. [52] and modified according to the method of Fibrich et al. [53]. The reaction mixture contained 100mM Tris buffer (pH 8.0), 0.5M HCl; the test sample was serially diluted in DMSO to yield seven concentrations (100–0.78/mL) to which PPE (5mM) was added and incubated for 15 min, followed by the addition of N-succinyl-ala-ala-ala-p-nitroanilide (4mM). The change in the absorbance of the reaction mixture was measured for 15 min, from which the rate was obtained. One unit of elastastolytic activity is defined as the release of 1 µM of p-nitroaniline/min. Ursolic acid was included as the positive drug control, as well as a vehicle control containing 2% DMSO and two negative controls (one in which no enzyme was added and the other in which no substrate was added). An IC_50_, or concentration at which 50% of the enzyme activity is inhibited was calculated using GraphPad Prism 4 (GraphPad Software, San Diego, CA, USA).

### 3.8. Statistical Analysis

The antioxidant and anti-inflammatory data were analyzed using a one-way analysis of variance (ANOVA), followed by Student’s *t*-test, using an Excel statistical application (Microsoft 365, Microsoft Corporation, Redmond, WA, USA).

## 4. Conclusions

This study isolated and characterized six compounds, stigmasterol (**1**), phenanthrene derivative (**2**), β-sitosterol, dodecanoic acid (**4**), ethyl nonadecanoate (**5**) and ethyl stearate (**6**), from *C. sexangularis* leaf for the first time. The compounds showed considerable in vitro antioxidant, anti-inflammatory and anti-elastase properties, with stigmasterol exhibiting the best bioactivity. Thus, this report may serve as a justification for the ethnomedicinal use of the plant for local skin treatment. It may also serve to validate the biological role of steroids and fatty acid compounds in cosmeceutical formulations.

## Figures and Tables

**Figure 1 molecules-28-03434-f001:**
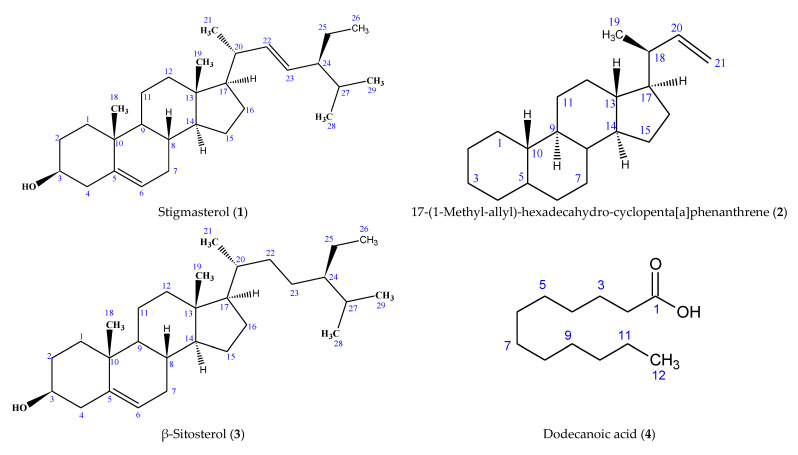
Structures of Compounds **1**–**6** Isolated from *Cyperus sexangularis* Leaf.

**Figure 2 molecules-28-03434-f002:**
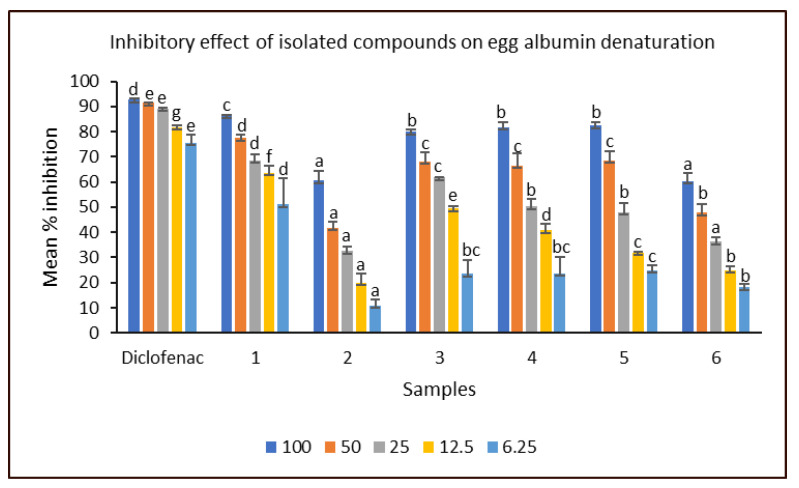
In vitro anti-inflammatory property of isolated compounds. Bioactivity expressed in terms of the ability of compounds to inhibit the denaturation of egg albumin; values in bar graph are expressed as mean ± SEM (*n* = 3). Isolated compounds (**1**–**6**): stigmasterol (**1**), phenanthrene derivative (**2**), β-sitosterol (**3**), dodecanoic acid (**4**), ethyl nonadecanoate (**5**) and ethyl stearate (**6**); diclofenac (standard); compounds were analyzed across the various concentrations (100–6.25 µg/mL) by one-way ANOVA; different alphabets on the same concentration (color) bar are considered significant at *p* < 0.05.

**Table 1 molecules-28-03434-t001:** ^1^H- and ^13^C NMR Spectral Data Assignment of Compounds **1** and **3** Isolated from *C. sexangularis* Leaf.

Position	Type of Carbon	Compound 1		Compound 3		Stigmasterol [25]
		^1^H NMR (Multiplicity, *J*)	^13^C NMR	^1^H NMR (Multiplicity, *J*)	^13^C NMR	^13^C NMR
1	CH_2_		37.40		37.48	37.6
2	CH_2_		34.08		31.86	32.1
3	CH	3.53 (1H, dt, *J* = 6.0 Hz)	71.95	3.52 (1H, tdd, *J* = 9.6, 6.4, 6.0 Hz)	71.78	72.1
4	CH_2_	2.25 (2H, d, *J* = 10.9 Hz)	42.42	2.25 (2H, d, *J* = 6.8 Hz)	42.28	42.4
5	C=CQ		140.89		140.66	141.1
6	HC=C	5.35 (1H, bd, *J* = 5.1 Hz)	121.87	5.32 (1H, bd, *J* = 5.2 Hz)	121.39	121.8
7	CH_2_		31.78		30.36	31.8
8	CH		32.04		29.73	31.8
9	CH		50.27		50.02	50.2
10	CQ		36.29		36.86	36.6
11	CH_2_		21.23		22.93	21.5
12	CH_2_		39.92		39.52	39.9
13	CQ		42.45		42.28	42.4
14	CH		56.91		56.60	56.8
15	CH_2_		24.45		28.09	24.4
16	CH_2_		29.29		29.73	29.3
17	CH		56.20		55.82	56.2
18	CH_3_	1.00 (3H, s)	12.12	0.99 (3H, s)	11.84	12.2
19	CH_3_	0.68 (3H, s)	18.93	0.66 (3H, s)	17.98	18.9
20	CH	2.02 (1H, m)	40.63	2.02 (1H, m)	36.24	40.6
21	CH_3_	0.92 (3H, d, *J* = 6.7 Hz)	19.97	0.91 (3H, d, *J* = 6.4 Hz)	19.23	21.7
22	HC=C	5.15 (1H, dd)	138.46		35.61	138.7
23	C=CH	5.01 (1H, dd)	129.41		25.74	129.6
24	CH		45.97		45.71	46.1
25	CH_2_		26.21		24.33	25.4
26	CH_3_	0.84 (3H, t, *J* = 8.1 Hz)	12.00	0.82 (3H, t, *J* = 4.5 Hz)	11.96	12.1
27	CH		29.85		29.69	29.6
28	CH_3_	0.82 (3H, d, *J* = 6.9 Hz)	19.54	0.80 (3H, d, *J* = 4.0 Hz)	20.88	20.2
29	CH_3_	0.80 (3H, d, *J* = 6.9 Hz)	19.18	0.78 (3H, d, *J* = 4.0 Hz)	19.28	19.8

Key: The chemical shifts of proton and carbon NMR signals are represented in parts per million (ppm) in deuterated chloroform; multiplicity: s, d, dd, dt, m, bs, tdd and t stand for a singlet, doublet, doublet of a doublet, doublet of a triplet, multiplet, broad singlet, triplet of a doublet of a doublet, and triplet, respectively; coupling constant (*J*) is expressed in Hertz (Hz); methyl (CH_3_); methylene (CH_2_); methine (CH); quaternary carbon (CQ).

**Table 2 molecules-28-03434-t002:** ^1^H- and ^13^C NMR Spectral Data Assignment of Compound **2** Isolated from *C. sexangularis* Leaf.

Position	Type of Carbon	Chemical Shift δ (ppm)		5β-Pregnane [27]	
		^1^H NMR (Multiplicity, *J*)	^13^C NMR	^1^H NMR	^13^C NMR
1	CH_2_	1.21–1.27 (m)	28.74	1.20–1.30 (m)	37.29
2	CH_2_	1.27–1.37 (m)	26.45	1.25–1.35 (m)	24.47
3	CH_2_	1.27–1.37 (m)	24.18	1.20–1.30 (m)	29.25
4	CH_2_	1.21–1.27 (m)	29.40	1.20–1.30 (m)	29.45
5	CH		31.70		25.20
6	CH_2_	1.27–1.37 (m)	29.07	1.20–1.30 (m)	29.70
7	CH_2_	1.27–1.37 (m)	25.83	1.20–1.30 (m)	22.73
8	CH		31.39		32.68
9	CH		29.74		27.98
10	CH		32.33		36.50
11	CH_2_	1.27–1.37 (m)	28.42	1.20–1.30 (m)	22.63
12	CH_2_	1.27–1.37 (m)	26.79	1.20–1.30 (m)	39.37
13	CH		32.96		42.40
14	CH		33.36		33.66
15	CH_2_		22.24		25.20
16	CH_2_		19.26		29.93
17	CH	1.55 (m, 1H)	36.88	1.6 (m, 1H)	31.93
18	CH	2.33 (m, 1H)	32.02	1.28 (m, 2H)	21.40
19	CH_3_	0.87 (d, 3H, *J* = 4.0 Hz)	13.81	0.85 (t, 3H)	19.75
20	CH	5.82 (m, 1H)	138.72	0.98 (s, 3H)	24.87
21	CH_2_	4.96 (d, 2H, *J* = 16.0 Hz)	113.65	1.10 (s, 3H)	14.12

Key: doublet (d); doublet of a doublet (dd); triplet (t); multiplet (m); methyl (CH_3_); methylene (CH_2_); methine (CH), coupling constant (*J*) is expressed in Hertz (Hz).

**Table 3 molecules-28-03434-t003:** ^1^H and ^13^C NMR Spectral Data Assignment of Compound **4** from *C. sexangularis* leaf.

Position	Type of Carbon	Compound 4		Lauric Acid [29]	
		^1^H NMR (Multiplicity, *J*)	^13^C NMR	^1^H NMR (Multiplicity, *J*)	^13^C NMR
1	CQ		179.84		180.0
2	CH_2_	2.31 (t, *J* = 10.0 Hz)	31.92	2.35 (t, *J* = 7.5 Hz)	34.0
3	CH_2_	1.56 (m)	29.36	1.63 (quintet, *J* = 7.5 Hz)	31.9
4	CH_2_	1.23–1.28 (m)	29.85	1.20–1.40	29.6
5	CH_2_	1.23–1.28 (m)	29.85	1.20–1.40	29.4
6	CH_2_	1.23–1.28 (m)	29.85	1.20–1.40	29.3
7	CH_2_	1.23–1.28 (m)	29.85	1.20–1.40	29.2
8	CH_2_	1.23–1.28 (m)	29.85	1.20–1.40 (m)	29.1
9	CH_2_	1.23–1.28 (m)	29.85	1.20–1.40 (m)	24.7
10	CH_2_	1.23–1.28 (m)	29.85	1.20–1.40 (m)	22.1
11	CH_2_	1.26–1.32 (m)	22.89	0.88 (t, *J* = 6.9 Hz)	14.1
12	CH_3_	0.86 (t, *J* = 8.0, 8.0 Hz)	14.12		

Key: Singlet (s); doublet (d); doublet of a doublet (dd); triplet (t); triplet of doublet of a doublet (tdd); multiplet (m); methyl (CH_3_); methylene (CH_2_); methine (CH); quaternary carbon (CQ); coupling constant (*J*) is expressed in Hertz (Hz).

**Table 4 molecules-28-03434-t004:** ^1^H- and ^13^C NMR Spectral Data Assignment of Compounds **5** and **6** Isolated from *C. sexangularis* Leaf.

Position	Type of Carbon	Compound 5		Compound 6		Ethyl Stearate [31] ^13^C NMR
		^1^H NMR (Multiplicity, *J*)	^13^C NMR	^1^H NMR (Multiplicity, *J*)	^13^C NMR	
1	C=O, ester		173.01		173.01	174.1
	O-CH_2_	4.07 (q, *J* = 6.7 Hz)	64.31	4.08 (q, *J* = 6.7, 6.7 Hz)	64.31	60.3
	CH_3_	1.26	14.30	1.26 (t, *J* = 21.6 Hz)	14.29	14.2
2	CH_2_	2.21 (t, *J* = 7.4 Hz)	34.59	2.21 (t, *J* = 7.4 Hz)	34.59	34.5
3	CH_2_	1.61 (m)	25.49	1.56–1.62 (m)	25.49	25.1
4	CH_2_	1.24 (m)	26.41	1.24–1.30 (m)	26.41	29.4
5	CH_2_	1.24–1.32 (m)	29.27	1.46–1.52 (m)	29.27	29.5
6	CH_2_	1.24–1.32 (m)	29.60	1.46–1.52 (m)	29.59	29.7
7	CH_2_	1.24–1.32 (m)	29.69	1.46–1.52 (m)	29.69	29.7
8	CH_2_	1.24–1.32 (m)	29.74	1.46–1.52 (m)	29.73	29.7
9	CH_2_	1.24–1.32 (m)	29.95	1.22–1.32 (m)	29.95	29.7
10	CH_2_	1.24–1.32 (m)	29.82	1.46–1.52 (m)	29.81	29.7
11	CH_2_	1.24–1.32 (m)	30.05	1.22–1.32 (m)	30.05	29.7
12	CH_2_	1.24–1.32 (m)	30.00	1.22–1.32 (m)	29.99	29.7
13	CH_2_	1.24–1.32 (m)	30.13	1.22–1.32 (m)	30.14	29.7
14	CH_2_	1.24–1.32 (m)	30.09	1.22–1.32 (m)	30.08	29.6
15	CH_2_	1.24–1.32 (m)	30.15	1.22–1.32 (m)	30.20	29.2
16	CH_2_	1.24–1.32 (m)	30.21	1.26–1.30 (m)	32.35	32.0
17	CH_2_	1.30–1.36 (m)	32.36	1.24–1.28 (m)	23.10	22.8
18	CH_2_	1.26–1.30 (m)	23.10	0.91 (t, J = 6.8 Hz)	14.29	14.2
19	CH_3_	0.91 (t, *J* = 6.9 Hz)	14.30			

Key: Singlet (s); doublet (d); doublet of a doublet (dd); triplet (t); triplet of doublet of a doublet (tdd); multiplet (m); methyl (CH_3_); methylene (CH_2_); methine (CH); coupling constant (J) is expressed in Hertz (Hz).

**Table 5 molecules-28-03434-t005:** Antioxidant Activities of Isolated Compounds from *Cyperus sexangularis* Leaf.

Isolated Compound	DPPH (IC_50_ ± SEM) (µg/mL)	NO (IC_50_ ± SEM) (µg/mL)	FRAP (µgAAE/mg)
**1**	38.18 ± 2.30 ^bc^	68.56 ± 4.03 ^bc^	303.58 ± 10.33 ^c^
**2**	57.63 ± 1.62 ^e^	105.73 ± 7.48 ^d^	196.21 ± 13.79 ^a^
**3**	40.60 ± 1.33 ^c^	72.48 ± 3.05 ^c^	257.88 ± 26.13 ^b^
**4**	35.99 ± 2.13 ^b^	65.04 ± 3.20 ^b^	245.94 ± 22.81 ^b^
**5**	43.82 ± 1.44 ^c^	69.74 ± 3.06 ^bc^	243.40 ± 19.72 ^b^
**6**	49.86 ± 2.29 ^d^	95.99 ± 4.93 ^d^	190.92 ± 11.68 ^a^
**ASC**	8.92 ± 1.26 ^a^	21.89 ± 1.32 ^a^	N/A

Footnote: Data are expressed as mean ± SEM (*n* = 3); data with different alphabets in superscripts are considered significant (*p* < 0.05), while those with same alphabets are comparable (*p* > 0.05); isolated compounds (**1**–**6**): stigmasterol (**1**), phenanthrene derivative (**2**), β-sitosterol (**3**), dodecanoic acid (**4**), ethyl nonadecanoate (**5**), ethyl stearate (**6**); L-ascorbic acid (**ASC**, positive control); Not applicable (N/A).

**Table 6 molecules-28-03434-t006:** Anti-elastase Activity of Isolated Compounds Isolated from *Cyperus sexangularis* Leaf.

Isolated Compound	Anti-Elastase Activity (IC_50_ ± SD) (µg/mL)
**1**	≥50 ^b^
**2**	≥100 ^c^
**3**	≥50 ^b^
**4**	≥50 ^b^
**5**	≥50 ^b^
**6**	≥50 ^b^
**Ursolic acid**	24.80 ± 2.60 ^a^
**E**	≥400 ^d^

Footnote: Isolated compounds (**1**–**6**): stigmasterol (**1**), phenanthrene derivative (**2**), β-sitosterol (**3**), dodecanoic acid (**4**), ethyl nonadecanoate (**5**), ethyl stearate (**6**); E-extracts (n-hexane, DCM, EtOAc, EtOH) exhibited ≥ 400 µg/mL anti-elastase activity; fifty percent inhibitory concentration (IC_50_); SD: Standard Deviation (*n* = 3); data with different alphabets in superscript are considered significant at *p* < 0.05.

## Data Availability

Not applicable.

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
