# Peer review of "Steroids and Fatty Acid Esters from Cyperus sexangularis Leaf and Their Antioxidant, Anti-Inflammatory and Anti-Elastase Properties"

_molecules, 2023, doi:10.3390/molecules28083434_

Round 1

Reviewer 1 Report

This article is devoted to obtaining and researching extracts of Cyperus sexangularis. The article provides a sufficient amount of experimental data, the article is adequately written. Obtaining extracts from plant biomass is an important topic in the chemistry of natural substances. So this work is in the trend of research in this area. I recommend that the authors improve the following points:

1. It is not clear from the text what yield of products was obtained during extraction. Specify it.

2. When describing experimental data, it is desirable to compare the results more with literature sources.

3. What is the main innovation in this work?

4. Please cite: 10.3390/molecules27186129.

5. It is advisable to double-check the quality of the English language.

6. Conclusions need to be substantially expanded.

Author Response

RESPONSE TO REVIEWER 1

Point 1: It is not clear from the text what yield of products was obtained during extraction. Specify it.

Response 1: The % yields of the afforded extracts have now been stated. Please check section 4.3 (MATERIALS AND METHODS - Gradient Extraction)

Point 2: When describing experimental data, it is desirable to compare the results more with literature sources.

Response 2: Experimental data have now been compared with literature sources. Please check section 2.3 (RESULTS AND DISCUSSION - Evaluation of biological activities of isolated compounds). 

Point 3: What is the main innovation in this work?

Response 3: The main innovation in this work is that the plant Cyperus sexangularis, though used domestically for mat making and ethnomedicinally for skin care in Southern Africa, there is dearth of information on the constituents of the plants and their biological properties. Thus, we isolated and identified three steroidal compounds (1-3), Dodecanoic (fatty) acid, and two fatty acid esters for the first time from the plant. Also, we have now generated some biological data (antioxidant, anti-inflammatory and anti-elastase activities) on the six compounds.

Point 4: Please cite: 10.3390/molecules27186129

Response 4: The referred article has now been cited in the manuscript.

Point 5: It is advisable to double-check the quality of the English language.

Response 5: The quality of the English language used in the manuscript has been double-checked.

Point 6: Conclusions need to be substantially expanded.

Response 6: Conclusion has been substantially expanded.

Thank you.

Reviewer 2 Report

The authors developed a study approaching the Chemical Composition, Antioxidant and Anti-Elastase Activities of Cyperus sexangularis Leaf Extracts. Furthermore, they isolated six natural product compounds and elucidated their chemical structures. The research is interesting and has scientific relevance, but some explanations must be made:

- Why weren't compounds tested?

- If possible, introduce the purity of isolated compounds.

- Introduce a table showing COSY correlations.

- Add mass and NMR spectra at least as supplemental material.

- Standardize the formatting for the names of the compounds in figure 1.

- Standardize data and table formatting. For example, some chemical shift and multiplicity for 5β-Pregnane are not shown. Add a character, if there is no experimental or literature signal.

- What's the meaning of " - " in table 1? Improve the subtitle of figures and tables.

- Figure 1: Show the stereochemistry at each chiral center of the compounds.

Author Response

RESPONSE TO REVIEWER 2

Point 1: Why weren't compounds tested?

Response 1: Compounds have now been tested and biological data on compounds have been added.

Point 2: If possible, introduce the purity of isolated compounds.

Response 2: Purity have been introduced in terms of the TLC profile and melting point range of compounds.

Point 3: Introduce a table showing COSY correlations.

Response 3: This increases the number of columns already created. One of the reviewers advised that the discussion on compounds be moderated since they are known compounds. Therefore, since few COSY correlations are observed on the spectra, information on COSY is now included in section 2.1.

Point 4: Add mass and NMR spectra at least as supplemental material.

Response 4: Spectra have been added as supplementary materials.

Point 5: Standardize the formatting for the names of the compounds in figure 1.

Response 5: The names of the compounds have been standardized.

Point 6: Standardize data and table formatting. For example, some chemical shift and multiplicity for 5β-Pregnane are not shown. Add a character, if there is no experimental or literature signal.

Response 6: Data and table formatting have been standardized. For conformity, the chemical shift and multiplicity for 5β-Pregnane are now included.

Point 7: What's the meaning of " - " in table 1? Improve the subtitle of figures and tables.

Response 7: The "-" implies that there is no proton signal on the 1HNMR spectrum because it is a quaternary carbon. However, all the "-" have been removed throughout the manuscript.

Point 8: Figure 1: Show the stereochemistry at each chiral center of the compounds.

Response 8: The stereochemistry of the compounds is now indicated.

Thank you.

Reviewer 3 Report

The manuscript Chemical Composition, Antioxidant and Anti-Elastase Activities of Cyperus sexangularis Leaf Extracts provides new insights to the Cyperus sexangularius, the plant species that is insufficiently researched. The scientific data about this species are very scarce, and the data provided within this manuscript represents novel and unpublished information.

Before publication of the article, it is necessary to correct some minor issues.

Line 30 and lines 127 and 134 – in the abstract isolated compounds 5 and 6 were named as “ethyl hexadecanoate (5) and ethyl stearate (6)”, but in the Results and discussion the names were not stated.

Line 262 – compound 5 was named as nonadecanoic acid ethyl ester and not ethyl hexadecanoate.

Please align the marks below the tables 1,2,3 and 4

Please pay attention to the chemical characterization of the compound 5.

Line 311 – Figure 1 -  The structure of compound 5 represents ethyl ester of nonadecanoic acid (C19), but it is named as ethyl-hexadecanoate.

Ethyl-hexadecanoate has molecular weight of 284.5 g/mol while nonadecanoic acid ethyl ester has molecular weight of 326.56 g/mol as you have concluded in the lines 258/259. The compound 5 was attributed to the molecular formula C21H42O2 (line 259) which also does not correspond to ethyl-hexadecanoate (C18H36O2). In the line 289 compound 5 was named as ethyl-hexadecanoate.

Line 321/322 – could you explain the claim “As shown in table 6, the four organic extracts of C. sexangularis demonstrated cytotoxicity against the HaCaT cells at ≥400 μg/mL concentration, which implies that the plant may contain some anti-elastase agents.“ How do data on cytotoxicity imply that extracts contain anti-elastase agents?

Line 324/325 – DCM extract demonstrated a cytotoxicity of 213.2±3.68 μg/mL against the HaCaT cells, but in the table IC50 for cytotoxicity is 160.45.

Line 326 – similarly as previous, for EtOAc extract IC50 is 160.45 while in the table 6. is 213.2 μg/mL.

Line 327 – it was concluded that the plant was considerably active with potential for skin care. How did you conclude this if all the IC50 values are higher than 400 μg/mL?

Attention should also be paid to the concentrations at which the extracts are cytotoxic. If cytotoxicity is achieved at concentrations lover than 400 μg/mL (DCM and EtOAc extracts) can you claim that these extracts have potential as anti-elastase agents? As I can see, they achieve anti-elastase activity at the concentrations higher than 400 μg/mL (the concentration at which they are cytotoxic).

Though from the literature it is well known that plant sterols have many bioactivities, I wonder why you did not perform antioxidant, anti-elastase, and cytotoxic tests on isolated compounds? This could explain their impact on the bioactivity of the extracts.

Line 472 – in the “Isolation of compounds” there is lack of data for compounds 5 and 6.

Please check the manuscript for spelling mistakes.

Author Response

RESPONSE TO REVIEWER 3

Point 1: Line 30 and lines 127 and 134 – in the abstract isolated compounds 5 and 6 were named as “ethyl hexadecanoate (5) and ethyl stearate (6)”, but in the Results and discussion the names were not stated.

Response 1: The error made has been corrected. The compound isolated is "Ethyl nonadecanoate".

Point 2: Line 262 – compound 5 was named as nonadecanoic acid ethyl ester and not ethyl hexadecanoate.

Response 2: The error made has been corrected. Compound 5 is "Ethyl nonadecanoate".

Point 3: Please align the marks below the tables 1,2,3 and 4

Response 3: Sorry, I do not understand what you are referring to.

Point 4: Please pay attention to the chemical characterization of the compound 5.

Line 311 – Figure 1 - The structure of compound 5 represents ethyl ester of nonadecanoic acid (C19), but it is named as ethyl-hexadecanoate.

Ethyl-hexadecanoate has molecular weight of 284.5 g/mol while nonadecanoic acid ethyl ester has molecular weight of 326.56 g/mol as you have concluded in the lines 258/259. The compound 5 was attributed to the molecular formula C21H42O(line 259) which also does not correspond to ethyl-hexadecanoate (C18H36O2). In the line 289 compound 5 was named as ethyl-hexadecanoate.

Response 4: Compound 5 has been reconciled to read ethyl nonadecanoate. the spectral data agrees with the elucidated structure, and it has been revised throughout the manuscript.

Point 5: Line 321/322 – could you explain the claim “As shown in table 6, the four organic extracts of C. sexangularis demonstrated cytotoxicity against the HaCaT cells at ≥400 μg/mL concentration, which implies that the plant may contain some anti-elastase agents.“ How do data on cytotoxicity imply that extracts contain anti-elastase agents?

Line 324/325 – DCM extract demonstrated a cytotoxicity of 213.2±3.68 μg/mL against the HaCaT cells, but in the table IC50 for cytotoxicity is 160.45.

Line 326 – similarly as previous, for EtOAc extract IC50 is 160.45 while in the table 6. is 213.2 μg/mL.

Line 327 – it was concluded that the plant was considerably active with potential for skin care. How did you conclude this if all the IC50 values are higher than 400 μg/mL?

Attention should also be paid to the concentrations at which the extracts are cytotoxic. If cytotoxicity is achieved at concentrations lover than 400 μg/mL (DCM and EtOAc extracts) can you claim that these extracts have potential as anti-elastase agents? As I can see, they achieve anti-elastase activity at the concentrations higher than 400 μg/mL (the concentration at which they are cytotoxic).

Though from the literature it is well known that plant sterols have many bioactivities, I wonder why you did not perform antioxidant, anti-elastase, and cytotoxic tests on isolated compounds? This could explain their impact on the bioactivity of the extracts.

Response 5: New biological data on the in vitro antioxidant, anti-inflammatory and anti-elastase activities have now been acquired on the isolated compounds, based on reviewers' request. As a result, the manuscript has been re-focused by reporting the isolation and characterization of the compounds as well as the afore-mentioned biological properties.

Point 6: Line 472 – in the “Isolation of compounds” there is lack of data for compounds 5 and 6.

Response 6: The isolation procedures for compounds 5 and 6 have been added.

Point 7: Please check the manuscript for spelling mistakes.

Response 7: The manuscript has been spell-checked. 

Thank you.

Reviewer 4 Report

The paper of Miya et al. “Chemical Composition, Antioxidant and Anti-Elastase Activities of Cyperus sexangularis Leaf Extracts” aimed to chemical and bioactivity study of Cyperus sexangularis Leaf Extracts. Generally, the paper contains flaws in logic reduced those small advantages that could be estimated as positive.

Highlights and strengths of the manuscript are:

Metabolome of Cyperus sexangularis is not well studied so six compounds isolated from hexane fraction of the extract were firstly described in species.  The results may further increase interest in Cyperus genus chemistry.  

Specific comments and suggested revisions:

- Despite the novelty of isolated compounds for the Cyperus sexangularis, all compounds are not new and the spectral data are not new too, so the precise description of spectral data is not necessarily. It will be enough to give references. Moreover, compounds are not unique and usual for many plants.

- The main problem in section 2.1 is why hexane fraction was used for chromatographic isolation. That might seem surprising, given the data about the lowest bioactivity of the fraction.

- Bioactivity section is routine and have no any scientific significance.

- The reference drawn up not in accordance with Instructions for Authors (https://www.mdpi.com/journal/molecules/instructions).

With all due respect to authors, I see no possibility to recommend paper for publication. The paper has serious flaws that can be corrected by adding new data and additional experiments.

Author Response

RESPONSE TO REVIEWER 4

Point 1: Despite the novelty of isolated compounds for the Cyperus sexangularis, all compounds are not new and the spectral data are not new too, so the precise description of spectral data is not necessarily. It will be enough to give references. Moreover, compounds are not unique and usual for many plants.

Response 1: Two other reviewers made some contributions that seem to allow the full description of the compounds to stay. Therefore, limiting or summarizing the discussion of the compounds might jeopardize their input and suggestion.

Point 2: The main problem in section 2.1 is why hexane fraction was used for chromatographic isolation. That might seem surprising, given the data about the lowest bioactivity of the fraction.

- Bioactivity section is routine and have no any scientific significance.

Response 2: New biological data have been acquired on the isolated compounds. This has helped to better re-focus the manuscript, by reporting on the isolation and characterization of six compounds from C. sexangularis plant for the first time, and determining their antioxidant, anti-inflammatory and anti-elastase properties. 

Point 3: The reference drawn up not in accordance with Instructions for Authors (https://www.mdpi.com/journal/molecules/instructions).

Response 3: The references have been revised in accordance with instructions for Authors.

Point 4: With all due respect to authors, I see no possibility to recommend paper for publication. The paper has serious flaws that can be corrected by adding new data and additional experiments.

Response 4: Additional experiments have been carried out and new biological data on the six isolated compounds have now been generated.

Thank you.

Round 2

Reviewer 1 Report

accepted

Author Response

RESPONSE TO REVIEWER 1

Manuscript has been revised accordingly.

Reviewer 2 Report

Based on their response to my previous review, the authors have adequately resolved the most of my concerns. Therefore, I endorse this manuscript for publication.

Some minor corrections:

  1. The authors need to format the paper according to the journal's guidelines.
  2. Table 1: Describe in the caption the meaning of "CQ"; "CH"; "CH2"; CH3".
  3. Table 3: Describe in the caption the meaning of "CQ". Correct "Compound Da".
  4. Table 4: Columns 5, 6, and 7 are not standardized. 1H and 13C: Numbers must be superscript.
  5. Number the figures in the supplementary material, e.g., Fig S1; Fig S2...
  6. Reference 52 is not listed in the references section. Please check all references and correct if necessary.

Author Response

REVIEWER 2 COMMENTS

Point 1: The authors need to format the paper according to the journal's guidelines.

Response 1: The paper has been formatted by following the journal's guidelines.

Point 2: Table 1: Describe in the caption the meaning of "CQ"; "CH"; "CH2"; CH3".

Response 2: The meaning of the terms in Table 1 have been described in the caption.

Point 3: Table 3: Describe in the caption the meaning of "CQ". Correct "Compound Da".

Response: The meaning of "CQ" in Table 3 has been described in the caption, while "Compound Da" has been corrected to read "Compound 4".

Point 4: Table 4: Columns 5, 6, and 7 are not standardized. 1H and 13C: Numbers must be superscript.

Response 4: Columns 5, 6, and 7 have now been standardized, while the numbers in 1H and 13C have been placed as superscripts.

Point 5: Number the figures in the supplementary material, e.g., Fig S1; Fig S2...

Response 5: The supplementary material has been numbered as Fig. S1 - Fig. S6.

Point 6: Reference 52 is not listed in the references section. Please check all references and correct if necessary.

Response 6: Reference number 52 has been corrected to indicate Chatterjee et al. (2012). However, an input from one of the reviewers has led to a slight change in the number of references. Therefore, the reference 52 in question is now "reference 50".

Thank you.

Reviewer 4 Report

After correction the paper of Miya et al. looks much better, but I continue to insist that there is no need for the spectral description of compounds in the manuscript. Moreover, the description of the Reference section is still does not meet the requirements. If the editor/s believes the paper deserves a publication, I am ready to support the decision.

Author Response

REVIEWER 4 COMMENTS

Point: After correction the paper of Miya et al. looks much better, but I continue to insist that there is no need for the spectral description of compounds in the manuscript. Moreover, the description of the Reference section is still does not meet the requirements. If the editor/s believes the paper deserves a publication, I am ready to support the decision.

Response: The spectral description of compounds has been expunged from the manuscript as suggested. The Reference section has been revised to meet the journal requirements.

Thank you.